Recent advances on the estimation of the thermal reaction norm for sex ratios

Abreu-Grobois F. Alberto 1
Morales-Mérida B. Alejandra 2 3 4
Hart Catherine E. 5 6
Guillon Jean-Michel 4
Godfrey Matthew H. 7 8 9
Navarro Erik 1
Girondot Marc marc.girondot@u-psud.fr 4
1 Laboratorio de Genética y Banco de Información sobre Tortugas Marinas (BITMAR), Unidad Académica Mazatlán, Instituto de Ciencias del Mar y Limnología, Universidad Nacional Autónoma de México , Mazatlán , Sinaloa , Mexico
2 Facultad de Ciencias Químicas y Farmacia, Universidad de San Carlos de Guatemala , Guatemala City , Guatemala
3 Doctorado en Ciencias Naturales para el Desarrollo (DOCINADE), Instituto Tecnológico de Costa Rica (TEC), Universidad Nacional (UNA), Universidad Estatal a Distancia (UNED) , San Jose , Costa Rica
4 Laboratoire Écologie, Systématique, Évolution, Université Paris Saclay, Centre National de la Recherche Scientifique, AgroParisTech , Orsay , France
5 Grupo Tortuguero de las Californias A.C. , La Paz , Baja California Sur , Mexico
6 Investigación, Capacitación y Soluciones Ambientas y Sociales A.C. , Tepic , Nayarit , Mexico
7 North Carolina Wildlife Resources Commission , Beaufort , NC , United States of America
8 Duke Marine Laboratory, Nicholas School of the Environment, Duke University , Beaufort , NC , United States of America
9 Department of Clinical Sciences, College of Veterinary Medicine, North Carolina State University , Raleigh , NC , United States of America
Lambert Max
Electronic publication date: 2020 Mar 4
Publication date: 2020
Volume: 8
Electronic Location ID: e8451
Received 2019 Sep 6; Accepted 2019 Dec 23
Copyright: ©2020 Abreu-Grobois et al.
Copyright year: 2020
Copyright holder: Abreu-Grobois et al.
License: This is an open access article distributed under the terms of the Creative Commons Attribution License, which permits unrestricted use, distribution, reproduction and adaptation in any medium and for any purpose provided that it is properly attributed. For attribution, the original author(s), title, publication source (PeerJ) and either DOI or URL of the article must be cited.
License URL: https://creativecommons.org/licenses/by/4.0/

Keywords: Temperature-dependent sex determination, Sex ratio, TSD, Reaction norm, Marine turtles, Incubation, Eggs, Olive ridley, Lepidochelys olivacea, Reptile

Funding: The authors received no funding for this work.

==============================
Temperature-dependent sex determination, or TSD, is a widespread phenomenon in reptiles. The shape of the relationship between constant incubation temperature and sex ratio defines the TSD pattern. The TSD pattern is considered a life-history parameter important for conservation because the wider the range of temperatures producing both sexes, the more resilient the species is to climate change impacts. We review the different published equations and methodologies that have been used to model TSD patterns. We describe a new flexible model that allows for an asymmetrical pattern around the pivotal temperature, which is the constant temperature producing both sexes in equal proportions. We show that Metropolis-Hastings with Markov chain produced by a Monte Carlo process has many advantages compared to maximum likelihood and is preferred. Finally, we apply the models to results from incubation experiments using eggs from the marine turtle Lepidochelys olivacea originating in Northeast Indian, East Pacific, and West Atlantic Regional Management Units (RMUs) and find large differences in pivotal temperatures but not in transitional ranges of temperatures.

Introduction

Sex determination is the biological process whereby an embryo can become male or female. Temperature-dependent sex determination (TSD), a special case of environmental sex determination (ESD), is widespread in the animal kingdom (Korpelainen, 1990) and is frequent in reptiles: all crocodilians have TSD, as do many turtles and some lepidosaurians (Valenzuela, 2004). In this system, the sexual phenotype of the embryo is defined by the temperature of the incubation occurring during a part of development termed the thermosensitive period (TSP) (Girondot, Monsinjon & Guillon, 2018b).

In oviparous reptiles, three patterns of TSD have been described, according to the changes in sex ratios as a function of different constant incubation temperatures (Bull, 1983; Ewert, Jackson & Nelson, 1994; Lang & Andrews, 1994). In the TSD Ia pattern, present in some turtles, low temperatures produce males and high temperatures produce females. The opposite occurs in some lizards (TSD Ib pattern). In the TSD II pattern, present in crocodile species, some turtles and some lizards, females are produced at low and high temperatures and more males are produced at intermediate temperatures. Temperature-dependent sex determination can be described as the reaction norm of the resulting sex ratio of embryos incubated at a range of constant temperatures. By definition, a reaction norm describes the pattern of the variation in phenotypic expression of a single genotype across a range of environments (Lewontin, 2000).

The mathematical relationship between incubation temperature and sex (or more specifically, the sex ratios produced by a suite of constant incubation temperatures), referred to as “thermal reaction norm”, is commonly characterized by two parameters: (1) the pivotal temperature (P), which is the constant temperature at which both sexes are produced in equal proportions (sex ratio = 1:1), and (2) the transitional range of temperatures (TRT), which is the range of constant temperatures that yields both sexes in variable proportions (Mrosovsky & Pieau, 1991). Note that there may be two values of P and TRT when considering the TSD II pattern.

The correct description of the thermal reaction norm for sex ratios (hereafter named TSD pattern) is not merely a game for biostatisticians. Indeed, variation in the TSD pattern can have profound implications for the conservation of TSD species, particularly in a world affected by climate change. For example, populations with a greater TRT should be more likely to evolve in response to new thermal conditions, thus putting them at lower risk to global change (Hulin et al., 2009; Hulin et al., 2008). When within- and among-population variation in the TSD patterns of 12 populations of painted turtles (Chrysemys picta) was studied, among-population variation in pivotal temperature could not be explained by geography or local thermal conditions, but the TRT was wider at lower latitudes, suggesting responsiveness to local incubation conditions (Carter et al., 2019). These results indicate that variation in TSD patterns among populations is not an artifact of incubation at constant temperatures and can provide insight into the ecology and evolution of temperature-dependent sex determination.

However, proper statistical tools are needed for robust analyses of the TSD patterns and for identifying the characteristics of interest. Although detailed methods for this have been developed and are freely available (Girondot, 2019a), their application has remained challenging, hampering authors with empirical data from fully capitalizing on their importance. Thus, the goal of this study is to provide a step-by-step workflow on how to analyze TSD patterns, focusing on the TSD Ia pattern, which is the most common TSD pattern found in turtles. We describe the most advanced statistical models to analyze the thermal reaction norm for the sex ratios produced by constant-temperature incubations, using data from published and unpublished research on the olive ridley marine turtle, Lepidochelys olivacea, as a test case. We choose this species for several reasons. It has a worldwide distribution, it is classified as Vulnerable by IUCN (Abreu-Grobois & Plotkin, 2008), and is often subjected to egg protection in beach hatcheries as a management strategy. Providing conservationists with adequate analytic tools to evaluate the sex ratios of resulting hatchlings (e.g., Dutton, Whitmore & Mrosovsky, 1985) is critical to reliably monitor the effect of temperature management actions required to counter climate warming. Furthermore, with a recent study indicating contrasting embryonic responses to incubation temperature from two rookeries (Mexico and Costa Rica) within the same Regional Management Unit (RMU) (Morales Mérida et al., 2015), analyses of TSD patterns are required to verify if regional differences exist.

Table 1 Constant temperature incubation data used in this study on TSD patterns in Lepidochelys olivacea.

Area	Country	RMU	Incubation temperature °C	Temperature amplitude °C	Males	Females	Intersexes	Reference	
Pirambu Beach	Brazil	West Atlantic	26.3	0.3	2	0	0	1	
Pirambu Beach	Brazil	West Atlantic	28.5	0.2	3	0	0	 	
Pirambu Beach	Brazil	West Atlantic	29.2	0.2	8	1	0	 	
Pirambu Beach	Brazil	West Atlantic	29.9	0.5	7	0	0	 	
Pirambu Beach	Brazil	West Atlantic	30.4	0.2	5	2	0	 	
Pirambu Beach	Brazil	West Atlantic	31.2	1.9	0	1	0	 	
Pirambu Beach	Brazil	West Atlantic	31.8	1.2	0	3	0	 	
Pirambu Beach	Brazil	West Atlantic	32.1	0.4	0	6	0	 	
Pirambu Beach	Brazil	West Atlantic	32.2	0.4	0	2	0	 	
Pirambu Beach	Brazil	West Atlantic	32.4	1.9	0	1	0	 	
Pirambu Beach	Brazil	West Atlantic	32.9	0.6	0	5	0	 	
Pirambu Beach	Brazil	West Atlantic	33.1	0.3	0	4	0	 	
Odisha	India	Northeast Indian	26.5	 	3	0	0	2, 3	
Odisha	India	Northeast Indian	28	 	8	0	0	 	
Odisha	India	Northeast Indian	29.5	 	2	3	0	 	
Odisha	India	Northeast Indian	30	 	0	4	0	 	
Odisha	India	Northeast Indian	31	 	0	2	0	 	
Odisha	India	Northeast Indian	31.5	 	0	9	0	 	
Nancite	Costa Rica	East Pacific	25	 	23	0	0	4	
Nancite	Costa Rica	East Pacific	28	 	30	1	3	 	
Nancite	Costa Rica	East Pacific	30	 	12	8	5	 	
Nancite	Costa Rica	East Pacific	32	 	0	23	0	 	
Nancite	Costa Rica	East Pacific	27	0.5	15	0	0	5	
Nancite	Costa Rica	East Pacific	29.4	0.5	23	1	0	 	
Nancite	Costa Rica	East Pacific	30.4	0.5	16	3	0	 	
Nancite	Costa Rica	East Pacific	32	0.5	0	19	0	 	
La Escobilla	Mexico	East Pacific	27	 	25	0	0	6	
La Escobilla	Mexico	East Pacific	32	 	0	26	0	 	
La Escobilla	Mexico	East Pacific	27.5	 	15	0	0	7	
La Escobilla	Mexico	East Pacific	32	 	0	9	0	 	
Playa La Destiladeras	Mexico	East Pacific	27.61	0.79 *	3	0	4	8	
Playa La Destiladeras	Mexico	East Pacific	32.24	0.81 *	0	5	6	 	
Playa La Destiladeras	Mexico	East Pacific	28.62	0.86 *	2	0	0	 	
Playa La Destiladeras	Mexico	East Pacific	32.29	1.05 *	0	1	1	 	
El Verde Camacho	Mexico	East Pacific	24	 0.1	16	0	0	9	
El Verde Camacho	Mexico	East Pacific	26	 0.1	24	0	0		
El Verde Camacho	Mexico	East Pacific	28	 0.1	20	0	0	 	
El Verde Camacho	Mexico	East Pacific	30	 0.1	15	4	0	 	
El Verde Camacho	Mexico	East Pacific	32	 0.1	0	20	0	 	
El Verde Camacho	Mexico	East Pacific	34	 0.1	0	5	0	 	
Notes.

* Maximum amplitude of temperatures within the middle-third of the incubation is shown. References: 1—Castheloge et al. (2018), 2–Dimond (1985), 3—Mohanty-Hejmadi, Behra & Dimond (1985), 4—McCoy, Vogt & Censky (1983), 5—Wibbels, Rostal & Byles (1998), 6—Merchant-Larios et al. (1997), 7—Merchant-Larios, Villalpando-Fierro & Centeno-Urruiza (1989), 8—López Correa (2010), 9–Navarro Sánchez (2015).

Material and Methods

Biological sources of the data

Data from different studies that incubated eggs at constant temperatures were extracted from publications (available in the “databaseTSD” file, as part of the R package embryogrowth (Girondot, 2019a). New data from our own unpublished studies (Navarro Sánchez, 2015) were also included (Table 1).

The version of the database available in the package can be obtained using the R command DatabaseTSD$Version[1]. The version used here (from embryogrowth ver. 7.6.7) is the 2019-11-19 version with 670 records for 25 species or subspecies.

Lepidochelys olivacea data including geographic origin, incubation temperatures and their amplitude were retrieved from this file. The Regional Management Units (RMUs) of olive ridley marine turtles as defined in Wallace et al. (2010) were inferred from the geographic origins. Whenever available, a temperature correction factor (difference between the measured incubator temperature and true internal egg temperature) was also obtained from the database. This correction factor has been reported in some publications (e.g., Godfrey & Mrosovsky, 2006) and has been shown to be relevant particularly when the substrate contains humidity (Tezak, Sifuentes-Romero & Wyneken, 2018). However, as few papers measure or report this parameter, it can be ignored when comparing studies where some, or all, do not report it. An alternative is to use an average correction factor based on all studies, but doing so will not contribute statistical information and will only shift fitted P values proportional to the correction factor. In the current analysis, the correction factor was not used and we used only sex ratio data from eggs incubated in temperature-regulated chambers. Incubation.temperature.Amplitude and 2ndThird.Incubation.temperature.Amplitude columns from databaseTSD refer to a measure of the variability of temperatures during the whole experimental incubation and the middle-third of the incubation, respectively. Mrosovsky & Pieau (1991) define the thermosensitive period (TSP) for sex determination as the interval of time when a change of incubation temperature results in a change in resultant sex ratio. The TSP begins with the formation of the genital ridge at stage 21 (sensu Miller, 1985), which corresponds to the onset of the gonad formation, and ends at stage 26 (sensu Miller, 1985), when the gonadal formation is nearly completed. The TSP occurs during the middle-third of the incubation period when incubation temperature is constant. When incubation temperature fluctuates during development, the TSP can shift, and the exact delimitation of TSP must be inferred by taking into account the thermal reaction norm for embryo growth (Girondot & Kaska, 2014). As a consequence, under naturally variable incubation conditions, the TSP is not exactly located at the middle-third of the incubation period (Girondot, Monsinjon & Guillon, 2018b). For example, if the incubation temperature at the beginning of incubation is low, embryos will not grow until the temperature increases. Subsequently, the growth of the embryo will start, and the TSP will be shifted towards the end of the incubation, in some cases even after the middle-third of the incubation period. We recommend excluding incubation data from analysis if the recorded temperatures exhibit an amplitude >2 °C. Indeed, even short daily bursts of high temperatures during egg incubation can lead to an abnormally high proportion of females, as compared to the expected sex ratio based on the average temperature (Georges, 1989; Georges, Limpus & Stoutjesdijk, 1994). Because egg incubation in natural conditions can expose eggs to wide daily fluctuations of temperature, especially for the relatively shallow nests laid by the olive ridley marine turtle, field collected data should not be used in this analysis. Indeed, by definition, the TSD pattern is defined by data collected from constant temperature incubations, thus the TSP is assumed to occur during the middle third of incubation without taking into account thermal reaction norm of embryon growth. Overall, the use of mean incubation temperatures obtained from nests in field conditions to feed a TSD pattern model has been shown to produce inaccurate results (Fuentes et al., 2017).

A further consideration relates to the difference among the resolution, accuracy, and uncertainty for temperature data loggers. Resolution refers to the data logger’s level of specificity for temperature in its memory. For example, a resolution of 0.5 °C indicates that temperatures will be recorded and reported in 0.5 °C bins, even if the electronic chips can internally read temperatures with a better resolution. Accuracy is represented in the logger’s technical datasheet as a range (±x °C), with x representing how close an individual recorded data point is to the true value. The uncertainty is a measure of the quality of the data logger temperature recordings, considering the accuracy, the resolution, and the sampling rate. Data logger uncertainty is then defined by the 95% confidence interval of the average temperature during a specific time, recorded during set sampling period by a data logger with known accuracy and resolution (Girondot et al., 2018a). The uncertainty of the mean temperature recorded every hour for 10 days is much lower than the accuracy and resolution of the logger. For example, using typical field conditions, an iButton DS1921G-F5# with accuracy = 1 °C and resolution = 0.5 °C has an uncertainty value of 0.15 °C (Girondot et al., 2018a). Overall, while the amplitude of temperatures during incubation should be minimized as much as possible, the uncertainty of the average temperature is a less important issue.

The data for this study originated from eggs collected in 6 nesting beaches (Pirambu, Brazil; Odisha, India; Playa Nancite, Costa Rica; El Verde Camacho, La Escobilla, and Playa La Destiladeras, Mexico) (Table 1, Fig. 1), belonging to 3 RMUs (West Atlantic, Northeast Indian, East Pacific). Though olive ridleys nesting on beaches in Pacific Mexico and Pacific Costa Rica belong to the same East Pacific RMU (Wallace et al., 2010), we analyzed these data separately because embryo growth dynamics from the two areas respond differently under a range of incubation temperatures (Morales Mérida et al., 2015).

Figure 1 Map showing locations of data collection.

(1) Pirambu Beach, Sergipe State, Northeastern Brazil, (2) Odisha (formerly spelled as Orissa), India, (3) Nancite, Costa Rica, (4) La Escobilla, Mexico, (5) Playa La Destiladeras, Mexico, (6) El Verde Camacho, Sinaloa, Mexico.

Data included 40 incubations at various constant temperatures with a total of 464 sexed embryos (277 males, 168 females, and 19 intersexes). The 19 intersexes were reported from only 2 studies out of 8, from incubation temperatures ranging from 27.6 to 32.3 °C. An intersex is a transient state during development when the gonads are ovotestes that exhibit characteristics of both testes and ovaries (Pieau & Dorizzi, 2004). After hatching, ovotestes generally evolve as normal testes (Pieau et al., 1998). Some adult gonads retain traces of intersexual characters when some oocytes may persist at the surface of testis in some species, whereas for others, no signs of intersex at the adult stage are observable. Because the criteria to define a gonad as an ovotestis are not entirely objective, we excluded data from hatchlings that were classified as intersex (n = 19 values, or 3.9% of all sexed turtles, Table 1).

The East Pacific RMU was overrepresented in the database (384 embryos, 190 from Mexico and 197 from Costa Rica) as compared to Northeast India (31 embryos) and West Atlantic (50 embryos). The number of incubation temperatures that produced mixed sex ratios was 2 for West Atlantic, 1 for Northeast Indian, 4 for East Pacific (Costa Rica), and 1 for East Pacific (Mexico).

Where stated, amplitudes of incubation temperatures were mostly <1 °C. Only two incubation temperatures in the West Atlantic RMU had amplitude >1.5 °C, but both included only 1 embryo and so are unlikely to have biased the results.

Thermal reaction norm for sex ratio

Several models have been published to describe the mathematical relationship between constant incubation temperatures and sex ratios. We enlist the most useful, with relevant comments:

• Logistic model. It is based on an equation originally developed to model population growth (Verhulst, 1838; Verhulst, 1846). This was the first model applied to constant incubation data (Girondot, 1999) and fitted using maximum likelihood with software TSD that is no longer recommended. The logistic model uses 2 parameters: P is the pivotal temperature, and S is one fourth of the inverse of the slope at P. The TRT can be easily calculated as TRTl = |S⋅Kl| with Kl being a constant equal to [2⋅ln(l∕1 − l)] with l being the limits to define the TRT. Girondot (1999) used l = 0.05 and then TRT was defined as the range of incubation temperatures with resulting sex ratios from 5% to 95%.

• Hill model. This model is used in biochemistry and pharmacology to reflect the binding of ligands to macromolecules as a function of the ligand concentration (Hill, 1910). From a mathematical point of view it is similar to a logistic equation with the natural logarithms of temperatures on the x-axis. The Hill model is therefore asymmetrical and uses 2 parameters. Control of the asymmetry in the shape is not possible. The Hill model was used previously to describe the TSD pattern but subsequently discarded for lack of sufficient flexibility (Godfrey, Delmas & Girondot, 2003).

• A-logistic model (A- for Asymmetrical) was specially developed for TSD pattern analysis in Godfrey, Delmas & Girondot (2003). It is based on the logistic equation with an additional parameter named K that controls the asymmetry. This K parameter is not the same as the Kl parameter used to calculate TRT for the logistic model (a mistake made by Carter et al., 2019). Godfrey, Delmas & Girondot (2003) also provided an equation to calculate the TRT. This model is asymmetrical, but the transitions towards the lower and upper asymptotes are not independent since they are controlled by a single parameter, K. The model has three parameters.

• Hulin model. Hulin et al. (2009), recognizing that the A-logistic model was insufficiently flexible in the transitions toward lower and upper asymptotes, introduced a modification to K, making it a linear function of temperature: K = K1 t + K2. Four parameters were therefore fitted, and TRT can only be calculated numerically. Unfortunately, this model is challenging to fit and often hangs on local minima because K can become very large during the search for maximum likelihood. In such situations, likelihood becomes insensitive to change in K1 or K2, and a local minimum is reached.

Additionally, it is noteworthy that the original description of TRT used the range of temperatures corresponding to sex ratios from 5% to 95%, which is unusual. Generally, the range is defined as occurring between 2.5% and 97.5%, thus encompassing a statistically meaningful 95% of the data. Sandoval, Gómez-Muñoz & Porta-Gándara (2017) questioned the use of the 5–95% limits, arguing instead that the TRT limit should be proportional to the number of eggs. This, however, is a misconception of the role of models in biology. A model is not a means to replace data but rather to obtain a generalized description of a biological phenomenon using a mathematical formula. In this case it needs to be framed within meaningful sex ratio limits. Thus, we recommend maintaining 5%–95% sex ratio as the limits of the TRT.

The lack of an ideal sigmoid model to describe TSD patterns (i.e., asymmetrical in the transitions toward lower and upper asymptotes) prompted us to develop a new, more versatile sigmoid function (see Supplemental Information 1) called flexible-logistic or flexit model: x<PS1=2K1−1SK12K1−1fx=1+2K1−1e4S1P−x−1∕K1x≥PS2=2K2−1SK22K2−1fx=1−1+2K2−1e4S2x−P−1∕K2

P is the pivotal temperature and S is the slope (first-order derivative) at P. K1 and K2 control the lower and upper asymptotes respectively (acute for positive values and obtuse for negative values).

TRT can be calculated exactly: TRT=14S2ln1∕1−lK2−12K2−1+14S1ln1∕1−lK1−12K1−1

A flexit model uses 4 parameters, a special case being K1 = K2 = 1, which is the logistic model. The model is not defined for K1 = 0 or K2 = 0. In this scenario, the corresponding value is replaced by 10−9.

The flexit model is included as a function in the HelpersMG R package (version 3.7 and higher) (Girondot, 2019b) and is included in the tsd() function of the embryogrowth R package (version 7.5 and higher) (Girondot, 2019a).

Overall, only two models are acceptable for our purpose: logistic and flexit. If an asymmetrical model is required, the Hill, A-logistic or Hulin models are not flexible enough when compared to a flexit version.

Fitting a model to the data: maximum likelihood

The fit of parameters (2 for logistic and 4 for flexit models) can be performed using the maximum likelihood methodology. The likelihood function (simply “likelihood”) expresses how probable a given set of observations is for different values of mathematical parameters. In the context of a model of TSD pattern, the observations are counts of categories of embryos. In most cases, two categories are found, males or females, and then a binomial distribution is used to estimate likelihood.

The likelihood of a set of M males and F females observed after an incubation t that produced a theoretical sex ratio of flexit (t; P, S, K1, K2) is named L: L=M+FMflexitt;P,S,K1,K2M1−flexitt;P,S,K1,K2F

with M+FM=M+F!M!F! being a constant for the set of observations.

An alternative that is described in some papers (e.g., Sandoval, Gómez-Muñoz & Porta-Gándara, 2017) is to fit the proportions with the hypothesis that proportions or their angular transformation are normally distributed. We do not recommend this method because it biased results: temperatures with fewer eggs incubated will have a larger than expected influence. Furthermore, even one sexed embryo at one incubation temperature provides useful information and should be included in the analysis.

It should be noted that sex ratio is etymologically referred to as M/F or F/M, but this definition is not practical in statistics. For practical purposes, we use sex ratio in terms of the relative frequency of males or females. The choice to work with male or female relative frequency has no importance and depends on the researcher’s preference. Here we use sex ratio as being relative male frequency, to be compatible with previous publications (Girondot, 1999; Godfrey, Delmas & Girondot, 2003; Hulin et al., 2009).

A possible alternative would be to incorporate the data on intersex hatchlings into the model, which tend to be more frequent at intermediate temperatures, and use a multinomial distribution. This possible solution has not yet been developed but could be an interesting avenue to explore in future studies for some species.

The likelihood value is often presented as its inverse natural logarithm (−ln L) for practical reasons: likelihoods are generally small numbers, thus −ln L will be positive numbers that are easier to manipulate.

The likelihood of a dataset of several incubation temperatures ti with Mi and Fi within a model is simply the product of the likelihoods for each temperature, Li, or the sum of the −ln Li.

To be able to fit the sex ratio thermal reaction norm using maximum likelihood, at least one incubation temperature producing a mixed sex ratio should be present in the study dataset. A rule of thumb is that fitting a model with p parameters necessitates at least p temperatures with mixed sex ratios. For example, if a logistic model (2 parameters) is fitted for a dataset with zero or only one temperature that produced a mixed-sex ratio, an infinite number of combinations of P and S will share the same likelihood. It is still possible to fit a logistic model to a dataset with only one temperature producing a mixed sex ratio, but standard error of parameters will be generally high. If a dataset has no mixed sex ratio, it is still possible to use a Bayesian model to describe the credibility interval of the parameters and the outputs (see below).

The estimate of parameters for TSD patterns using maximum likelihood serves various purposes: (i) It allows an estimate of the confidence interval of the outputs (see below), (ii) it provides an estimate of the quality of fit (see below), (iii) it facilitates the comparison across datasets because their fitted parameters can be compared even if incubation temperatures were different, (iv) it can be used as a prior for Bayesian analysis (see below), and (v) it can be used as a starting point for iterations using a Metropolis–Hastings algorithm with a Monte-Carlo Markov chain in Bayesian analyses (see below).

Standard error and confidence interval

The standard error is an important indicator of the precision of an estimate of a sample statistic for a population parameter. Its calculation is based on the Hessian matrix, which is the matrix of second-order partial derivatives of the likelihood for all pairwise parameters. The second-order derivative of a function at its maximum measures a more or less acute form of the function around the maximum. If a parameter is slightly shifted from its value at maximum likelihood and the likelihood changes drastically, this denotes a robust parameter estimate. At the same time, its standard error, which measures how well the parameter is known, will be low. On the contrary, if the likelihood is insensitive to changes of a parameter, it means the data do not provide information to fit this parameter and its standard error will be high. The inverse of the Hessian matrix is an estimator of the asymptotic covariance matrix. Hence, the square roots of the diagonal elements of a covariance matrix are estimators of the standard errors. A parameter obtained using maximum likelihood is normally distributed asymptotically.

This mathematical definition is sometimes problematic when the standard error is large and the effect of the parameter on the function changes drastically at some value. An example is the case for the S parameters because the TSD pattern is completely the reverse for -S (females at lower temperatures rather than high). Thus, if the standard error of S is large, at the ends of the S distribution the model will become completely reversed, and male production will be predicted at feminizing temperatures. The coefficient of variation for one parameter estimate is a standardized measure of dispersion with CV = SE∕mean. The larger the coefficient of variation, the worse the estimate of the parameter in the analysis.

The confidence interval for a parameter can be obtained directly from its point estimate and its standard error, assuming that it is normally distributed asymptotically. The confidence interval for a combination of parameters (for example, TRT) requires a more complicated calculation. Two main strategies are available to calculate the confidence intervals for a combination of parameters: the delta method and parameter resampling. The delta method uses the approximate probability distribution for a function of an asymptotically normal statistical estimator from knowledge of the limiting variance of that estimator. In short, if the standard error of the maximum likelihood parameters are known, the delta method permits an estimate of the distribution of any function of these parameters. The alternative is to generate many random numbers for the variance and covariance matrix of the estimators using the Cholesky decomposition (Tanabe & Sagae, 1992). The function of interest is then applied to each set of random numbers. The advantage of the delta method is its rapidity. However, the assumption of a normal distribution for an estimator is important. It precludes the use of the delta method on a truncated distribution such as the S distribution of a TSD pattern. If for example, S changes sign, the likelihood degrades so much that it would be considered a truncated distribution. The advantage of the generation of many random numbers from the Hessian matrix is that it is possible to check each set of numbers for coherence and discard some if necessary. In the case of models for the TSD pattern, when the confidence interval of sex ratio according to temperature or TRT is estimated, we ensure that S, K1 − 1, and K2 −1 do not change signs during resampling. However, this method will artificially reduce the confidence interval because the highly divergent values are removed. Using a Bayesian Metropolis–Hastings MCMC procedure solves this problem (see below).

Quality of fit

Often the quality of fit is measured by the determination coefficient R2, which is derived from comparisons between observations and predictions. Whereas the determination coefficient has meaning when the distribution of the dependent variable is normally distributed, in the case of univariate regression, it has no meaningful statistical properties when used with proportions. Thus, R2 should not be used in most of these cases and particularly here where we need to measure the fit of the model to sex ratio data. Instead, deviance can be used as a goodness-of-fit measure for a statistical model. It is twice the difference between the logarithm of likelihood of the saturated model (lnLS). In a saturated model, the fitted sex ratio is replaced by the observed sex ratio and the model fits the data perfectly, and the logarithm of maximum likelihood of the fitted model (lnLM) is: D=2lnLS− lnLM.

Deviance has an asymptotic χ2 distribution with the degrees of freedom calculated from the difference of the number of parameters in the saturated and the fitted model. Note, however, that if there are few observations (which is often the case), the distribution of deviance can deviate substantially from a χ2 distribution and the test result could be wrong. For this reason, we developed an additional deviance test by randomly generating null deviance distributions with the same characteristics as those from the observations (the same number of incubation temperatures and the same number of eggs per incubation temperature). The probability that the observed deviance is obtained with the experimental design is then calculated by comparing the observed deviance and the distribution of deviances under the null hypothesis.

Comparison of models: Akaike information criterion and Akaike weight

When several models are fitted to the same datasets of observations, the comparison of the performance of the different models can be assayed using AIC (Akaike Information Criterion) (Akaike, 1974). AIC is a measure of the quality of the fit, while it simultaneously penalizes for the number of parameters in the model. It facilitates the selection from a set of models the best compromise between fit quality and over-parametrization. AICj=−2lnLj+2pj

with Lj being the likelihood of the model and pj the number of parameters of the model j.

When a set of k models are tested, the model with lowest AIC is considered to be the best non-overparametrized fit. It is important to note that AIC value itself is not strictly informative in terms of absolute model fit.

A corrected version of AIC for small sample sizes, named AICc, has been proposed when the model is univariate and linear with normal residuals (Hurvich & Tsai, 1995): AICc=AIC+2pp+1n−p−1

The exact formula can be difficult to determine when these conditions are not met, in which case the previous formula could be used (Burnham & Anderson, 2002). In general, the AICc should always be used instead of AIC (Burnham & Anderson, 2002), especially for datasets comprised of small sample sizes, which is typical for sex ratio studies, and particularly when the study targets protected species.

When a set of models is compared, it is possible to estimate the relative probability that each model is the best among those tested using the Akaike weight (Burnham & Anderson, 2002): Akaikeweightj=e12AICcj−minAICc ∑i=1ke12AICci−minAICc

We use both AICc and Akaike weight to compare the fitted logistic and flexit models for our datasets.

The utility of model selection can be further extended to test for potential differences in the results from two or more datasets. In this case, the complete data are split into several subsets with each individual dataset being represented once and only once. The test question is: can the collection of datasets be better modeled with a single set of parameters or should each dataset be modeled with its own set? In this situation, Bayesian Information Criterion (or BIC) should be used instead of AIC or AICc, because the true model is obviously among the tested alternatives: BIC=−2lnL+plnn

When the BIC statistic is used, all the priors of the models tested are assumed to be identical. It is also possible to estimate BIC weights by replacing AICc with BIC in the Akaike weight formula. The w-value has been defined as the probability that several datasets can be correctly modeled by grouping instead of independently (Girondot & Guillon, 2018).

In our example, a model will be fitted first to the combined datasets and BIC (combined) will be estimated with p parameters. Then each dataset will be fitted separately and a set of k BIC (separate) values will be generated, each with p parameters, thus using a total of k.p parameters. This is similar to the case of fitted models with a dataset effect. The global -ln likelihood for the separate fits is simply the sum of the -ln likelihoods. Then the BIC weights will provide a statistically sound method of choosing between the hypotheses that a single model is sufficient for all datasets or that each dataset is best fitted with a different model. We named w- value the BIC weight and we propose this statistic as a replacement for the contentious p-value (Girondot & Guillon, 2018).

Bayesian approach using Metropolis–Hastings with Monte-Carlo Markov chain

The Metropolis–Hastings algorithm is a Markov chain Monte Carlo (MCMC) method for obtaining a sequence of random samples from a probability distribution (Hastings, 1970; Metropolis et al., 1953). This method is now widely used as it offers a high-performance tool to fit a model. To run a Bayesian analysis with this algorithm, several parameters must be defined for each estimator in the model. The following terms are those used in the function tsd_MHmcmc() of the R package embryogrowth:

1. Density: The R function for density distribution used for the prior. Generally, uniform or Gaussian distribution are used with dunif or dnorm, respectively.

2. Prior1 and Prior2: The parameters describing the prior distribution. For dunif, Prior1 and Prior2 are respectively the minimum and maximum of the distribution, and for dnorm, they are the mean and standard deviation of the distribution;

3. SDProp: The standard deviation for each new parameter;

4. Min and Max: The range of possible values defined by the minimum and maximum;

5. Init: An initial starting point for the Markov chain.

It is beyond the scope of this paper to fully explore the fine details of this algorithm, and rather we focus on how to use it.

The choice of the prior is not straightforward (Lemoine, 2019) and can be critical if only a few observations are available. A uniform distribution for priors indicates that all values within a range are equally probable, whereas a Gaussian distribution can use a mean and standard deviation obtained from a previous analysis with the same or other species.

At the end of the analysis, it is essential to evaluate the distribution of posteriors. If they are the same as the distribution of priors, it generally means that the observations did not help inform the fit for this parameter. In this case, it implies that the results are dependent on the priors and not on the data and, therefore, the results should not be used, or used with caution.

During the iteration process, a Markov Chain is constructed using the actual parameter values πt on which a new proposed random function defined by its standard deviation (s) is applied, πt+1=Nπt,s. This is the Monte-Carlo process. The standard deviation (s) for a new proposal is a compromise between the two constraints: if the values are too high, the new values could yield results far from the optimal solution, while if they are too low, the model can become stuck in local minima. The adaptive proposal distribution (Rosenthal, 2011) as implemented in R package HelpersMG (Girondot, 2019b) will ensure that the acceptance rate is close to 0.234, which is the optimal acceptance rate (Roberts & Rosenthal, 2001). The burn-in value is the number of iterations necessary to stabilize the likelihood. It can be low (around 10) when the starting values are the maximum likelihood estimators. The total number of iterations required is defined after an initial run of 10,000 iterations (Raftery & Lewis, 1992). The result of the MCMC analysis is a table with one set of values for the estimators at each iteration. The mean and standard deviation summary statistics can be calculated from this table using the coda R package (Plummer et al., 2011), or it is also possible to estimate quantiles. The use of quantiles has the advantage that it does not require any hypothesis on the output distribution; hence an asymmetric distribution is well accommodated. When the result of a MCMC analysis is used to estimate a function of these estimators (for example, the TRT), each individual set of parameters obtained during the MCMC search should be used to generate the posterior distribution of outputs and this distribution can be summarized using mean, standard deviation or quantiles. The standard deviation of the MCMC output is the standard error of the corresponding parameter.

We follow the ISO 80000-1:2009 standard indicating that numerical value and unit symbol are separated by a space, including the  °C symbol (ISO/TC 12 Quantities and units & IEC/TC 25 Quantities and units, 2009).

Results

Maximum likelihood estimates for grouped data

Logistic and flexit models were both fitted to the comprehensive dataset: 40 incubations with temperatures from 24 °C to 34 °C and 468 sexed embryos identified as males or females. Mixed-sex ratios were observed in embryos or hatchlings from 8 incubations (Table 1). Upon comparing both models (Fig. 2 and Table 2), the flexit model was selected based on AICc and Akaike weight as the latter suggested there was a 97% probability that the flexit model was the best. While the logistic model showed the typical symmetric shape, the flexit model showed a strong asymmetrical pattern: transition from all-male condition to the pivotal temperature is smooth, whereas the transition from the pivotal temperature to all-female production was abrupt (Fig. 2). As a consequence, the estimated pivotal and transitional range of temperatures obtained from both models were different (Fig. 2).

Figure 2 Logistic (A) and flexit (B) models fitted with all olive ridley incubation data.

Confidence intervals were estimated using 10,000 random numbers obtained from the Hessian matrix. The standard errors for the S, K1, and K2 parameters in the flexit model were high and during resampling some trials were removed as their signs had reversed (see text). As a consequence, the estimated confidence interval is biased towards a lower value. The dark grey zone is the TRT 5% and the light grey zone is the 95% confidence of the TRT. The points correspond to observations and the bars are their 95% confidence intervals for the sex ratios according to the Wilson method (1927). The plain curve shows the maximum likelihood model and its 95% confidence interval is shown as dashed lines. The pivotal temperature is indicated by the vertical dash-dotted line.

It is worthwhile noting that the standard error and therefore the coefficient of variation for the S, K1 and K2 parameters in the flexit model was large (S =  − 0.79 SE 0.3; K1 =−1.72 SE 1.59; K2 =200 SE 2485.61, see Supplemental Information). In other words, the flexit model did not precisely describe all parameters, even though it was the selected model.

The tests of the deviances indicated that the models fit the observations correctly: p = 0.16 and p = 0.56, respectively, for the logistic and flexit models on global data. The same is true for the fit at RMU or country scale except for Costa Rica data (p = 0.01, Table 3). The distribution of 1000 estimates of deviance for both the logistic and the flexit models is shown in Tables 2 and 3. All these tests gave a probability of more than 0.05, indicating that the models fit the observations relatively well. From these observations, we recommend against the use of χ2 approximations to test the degree to which the models fit the data and, instead, use comparisons based on generated null distributions. Based on this methodology, the function tsd() of the phenology R package returns the probability that the observations fit the null model in the object p.Deviance.Null.model (Girondot, 2019a).

Finally, because some parameters of the flexit model are nearly impossible to fit with these datasets (coefficient of variation for K1 = 0.92, and for K2 = 12.43), the flexit model is not applied further in this paper (see Supplemental Information 2 for a Bayesian MCMC with flexit model).

Comparison of the maximum likelihood estimates for RMU data

The characteristics of the fitted logistic model for the 3 RMUs and Mexico and Costa Rica within the East Pacific RMU are shown in Table 3. The fitted S values for East Pacific (Mexico) and Northeast India observations have substantial standard errors and confidence intervals. These two datasets have only one mixed-sex ratio, and the maximum likelihood approximation failed to provide a correct estimate of the S parameter.

The analysis shows that, within the East Pacific RMU, a single TSD pattern is sufficient to model the combined Mexico and Costa Rica data (w-value = 0.87), whereas there is a 0.97 probability (w-value) that at a worldwide scale there are differences in TSD parameters for the different olive ridley populations (Table 4).

Table 2 Summary of logistic and flexit model tested for Lepidochelys olivacea incubation data (East Pacic, West Atlantic, and Northeast Indian).

“p-random” column is to the probability that the observed deviance was obtained using a random sampling with the same characteristics (same number of temperatures and eggs per temperature and sex ratio probability obtained from the tted model). Line with bold text indicates the best model.

Model	P (mean SE)	S (mean SE)	K1 (mean SE)	K2 (mean SE)	−Ln L	AICc	ΔAICc	Akaike Weight	Deviance	df	p-value	p-random	
Logistic	30.39 SE 0.09	−0.41 SE 0.05			32.6	69.52	6.92	0.03	45.68	38	0.183117	0.08	
Flexit	30.57 SE 0.11	−0.79 SE 0.3	−1.72 SE 1.59	200 SE 2477.58	26.73	62.6	0	0.97	33.95	36	0.566649	0.22	

Table 3 Characteristics of logistic models tested for Lepidochelys olivacea incubation data (East Pacic, West Atlantic, and Northeast Indian).

“p-random” column is to the probability that the observed deviance was obtained using a random sampling with the same characteristics (same number of temperatures and eggs per temperature and sex ratio probability obtained from the fitted model). Cells with large coefficient of variation are in bold and could produce a sign change for the parameter during resampling.

RMU	Country	P (mean SE)	S (mean SE)	−Ln L	Deviance	df	p-value	p-random	
Northeast India	India	29.49 SE 0.33	−0.03 SE 0.81	1.06	0	4	1	0.4	
West Atlantic	Brazil	30.63 SE 0.23	−0.36 SE 0.12	4.46	4.76	10	0.90	0.56	
East Pacific	Costa Rica + Mexico	30.46 SE 0.1	−0.37 SE 0.05	16.32	19.43	20	0.49	0.28	
”	Costa Rica	30.5 SE 0.13	−0.42 SE 0.07	13.5	16.83	6	0.01	0.26	
”	Mexico	30.16 SE 1.9	−0.12 SE 1.43	1.52	0	12	1	0.41	

Table 4 Comparison using BIC and BIC weight (or w-value) of the homogeneity of TSD pattern.

(A) Within the East Pacic RMU and (B) at a global scale. Selected models are indicated in bold font.

A : Comparison within East Pacific RMU	
Series	BIC	ΔBIC	BIC weight	
All East Pacific grouped	38.06	0.00	0.87	
Mexico and Costa Rica separated	41.82	3.76	0.13	
B: Worldwide comparison	
World	71.99	7.32	0.03	
East Pacific, Northeast Indian, and West Atlantic separated	64.67	0.00	0.97	

Bayesian estimates

Priors for the Bayesian process were chosen with a Gaussian distribution, with an average set as the fitted parameters, which were estimated from the global maximum likelihood analysis (Fig. 2, Table 2), and a standard deviation large enough to avoid imposing too high a constraint on the output. Minimum and maximum values for P were 25 and 35 °C, respectively, with S spanning from −2 to 2 and Kx from −500 to +500. The standard deviation for the new proposal was initially chosen to be 2 for P and 0.5 for S, K 1, and K 2. They were adjusted during the iterations to have acceptance rates closer to 0.234. A total of 100,000 iterations were performed.

Though both the logistic and flexit models were fitted using Bayesian MCMC, only the results for the logistic will be discussed here because some parameters of the flexit model are nearly impossible to fit with these datasets (coefficient of variation for K2 = 12.43) (see Supplemental Information 2 for a Bayesian MCMC with flexit model).

The distribution of priors and posteriors for the parameters derived from the East Pacific analyses is shown in Figs. 3A and 3B. The distinctness of the posterior and prior distributions verify that the fit was guided primarily by the observations. The posterior distributions of the P and S parameters were relatively independent (Fig. 3C). Distribution of other posteriors is shown in Supplemental Information 2.

Figure 3 Results of the Bayesian MCMC analysis for East Pacific Lepidochelys olivacea RMU.

(A, B) Distribution of priors (plain line) and posteriors (histograms) for P (A) and S (B) for East Pacific RMU. The covariation of posteriors for P and S is shown in (C).

TSD pattern fits using maximum likelihood and Bayesian MCMC (Figs. 4 and 5) were derived for the 3 RMUs, and separately for Mexico and Costa Rica (East Pacific RMU) together with the 0.025, 0.5 and 0.975 quantiles (95% of values are located between 0.025, and 0.975 quantiles) for pivotal temperature and transitional range of temperatures (Table 5).

Figure 4 TSD patterns as modeled by a logistic function and fitted using maximum likelihood.

(A) East Pacific, (B) Mexico, (C) Costa Rica, (D) West Atlantic, (E) Northeast Indian. The dark grey zone is the TRT 5% and the light grey zone is the 95% confidence of the TRT. The points correspond to observations and the bars are their 95% confidence intervals. The plain curve shows the maximum likelihood model and its 95% confidence interval is shown as dashed lines. The pivotal temperature is indicated by the vertical dash-dotted line. Note that Mexico, East Pacific and Northeast Indian datasets each have only one temperature with mixed sex ratio.

Figure 5 TSD patterns modeled as a logistic function fitted using Bayesian MCMC.

(A) East Pacific, (B) Mexico, (C) Costa Rica, (D) West Atlantic, (E) Northeast Indian. The dark grey zone is the TRT 5% and the light grey zone is the 95% confidence of the TRT. The points correspond to observations and the bars are their 95% confidence intervals. The plain curve shows the maximum likelihood model and its 95% confidence interval is shown as dashed lines. The pivotal temperature is indicated by the vertical dash-dotted line. Note that Mexico, East Pacific and Northeast Indian datasets each have only one temperature with mixed sex ratio.

Table 5 Pivotal temperature and transitional range of temperatures (5%) for Lepidochelys olivacea RMUs.

Quantiles (0.025, 0.5, and 0.975) for pivotal temperature and transitional range of temperatures (5%) using maximum likelihood (MLE, upper line) and Bayesian MCMC (Bay. MCMC, lower line) with logistic and flexit model for TSD pattern.

		Pivotal temperature in °C	Transitional range of temperatures in °C	
Quantiles		0.025	0.5	0.975	0.025	0.5	0.975	
RMU	Logistic model	
East Pacific	MLE	30.27	30.46	30.66	1.66	2.19	2.77	
Bay. MCMC	30.26	30.46	30.66	1.72	2.24	2.89	
Northeast Indian	MLE	28.74	29.27	29.50	0.17	3.28	10.89	
Bay. MCMC	28.72	29.35	29.76	0.34	1.72	4.76	
West Atlantic	MLE	30.18	30.63	31.06	0.79	2.14	3.51	
Bay. MCMC	30.23	30.65	31.15	1.29	2.44	4.53	
RMU	Flexit model	
East Pacific	MLE	30.58	30.92	31.23	1.96	2.81	4.08	
Bay. MCMC	30.44	30.76	31.29	1.48	2.33	3.40	
Northeast Indian	MLE	29.66	34.31	45.71	na	na	na	
Bay. MCMC	29.00	29.43	29.74	0.37	0.87	4.65	
West Atlantic	MLE	30.12	30.36	30.56	na	na	na	
Bay. MCMC	30.34	30.82	31.52	1.02	2.37	28.13	

The plotted posterior distributions for P vs. TRT from the 2 RMUs and Mexico and Costa Rica separately, are shown in Fig. 6. While posterior values for P suggest two separate groups (Northeast India on one side, and East Pacific and West Atlantic on the other), the spread of TRT values does not suggest any differences.

Figure 6 Distribution of 100,000 posteriors for pivotal temperatures and transitional range of temperatures for the logistic model fitted using Bayesian MCMC.

A 20,000 random subsample of posteriors is shown. Ellipses including 75% of the points are drawn.

Discussion

After the discovery of temperature-dependent sex determination in reptiles (Charnier, 1966; Pieau, 1971; Pieau, 1972), the description of the TSD pattern provided a basis for its understanding and for comparisons among species (Bull, Vogt & McCoy, 1982; Ewert, Etchberger & Nelson, 2004; Ewert, Jackson & Nelson, 1994; Ewert, Lang & Nelson, 2005; Ewert & Nelson, 1991; Godfrey et al., 1999; Mrosovsky, 1988; Yntema & Mrosovsky, 1982). However, significant methodological advancements have been made since the original studies. For example, straight-line interpolations of the sex ratio values on each side of the 50% level were proposed by N. Mrosovsky (pers. comm., 1992) (see also Mrosovsky & Pieau, 1991). Limpus, Reed & Miller (1983) refined the approach using a graphical method (Litchfield & Wilcoxon, 1949) using a curve from the intercept and the slope of a straight line in the log dose vs. probit effect scale. This method allowed statistical comparisons between samples but required at least two values in the 16% to 84% dose–effect range, confidence limit calculations, and re-testing between samples. Starting in 1999, new statistical methods were developed (Girondot, 1999; Godfrey, Delmas & Girondot, 2003; Hulin et al., 2009) and the R package embryogrowth (Girondot, 2019a) has included the tsd() function since version 2.0.0 in 2013.

More recently, attempts to correlate TSD parameters with life-history parameters were successful (Carter et al., 2019; Hulin et al., 2009) but sometimes produced unexpected results. For example, when these were compared between populations in Chrysemys picta, P could not be explained by geography or local thermal conditions, but the TRT was wider at lower latitudes (Carter et al., 2019). An explanation for these difficulties could be that the models for the TSD pattern used did not correctly reflect the true TSD pattern.

The TSD pattern has also been used to estimate sex ratios from natural nests using the average nest temperature for the total incubation period, which can result in biased data because temperature determines sex only during the thermosensitive part of development. When the average temperature experienced during the absolute middle third of the incubation period is used, sex ratio is also potentially biased because the thermosensitive period for sex determination shifts depending on the temperature (Georges et al., 2005; Girondot, Monsinjon & Guillon, 2018b). Recently, improved models that take into account changes in the rate of embryonic development affected by variations in the incubation temperature have been developed (Massey et al., 2019; Monsinjon et al., 2019a; Monsinjon et al., 2019b). Promising results have been obtained, but further analyses of in-situ empirical data would benefit the field (Fuentes et al., 2017).

Nevertheless, the mathematical and statistical complications of currently available methods to analyze and study TSD patterns were a barrier for the full understanding and further analyses of sex ratios. The purpose of this article was to completely describe the procedures and the R code to familiarize readers with the analyses and the outputs.

The main conclusions of the example analysis are summarized here. A single model is sufficient to describe Mexican and Costa Rican rookeries within East Pacific RMU. This result is consistant with the phylogeography of this species (see Fig. 2 in Bowen et al., 1998). The pivotal temperature for the India rookery (Northeast Indian RMU) differs substantially from the values estimated for both Mexico and Costa Rica (East Pacific RMU), and also from the estimation for Brazil (West Atlantic RMU). However, no differences among the RMUs were observed from the estimated TRTs. The use of the Bayesian Metropolis–Hastings algorithm with Markov chains generated by a Monte-Carlo process provides a substantial improvement to the model fit as compared to the maximum likelihood fit, especially when few mixed-sex ratio results are available. The quality of the fit, as measured by a deviance test, was generally reliable, but in one case the generation of null distribution for deviance gave a result different from the χ2 approximation. We suggest that a null distributions of deviance are more reliable than χ2 approximations to test deviance and should be chosen when possible.

We hope that this methodological paper will be useful and encourage researchers to explore new hypotheses to understand the ecological and evolutionary significance of temperature-dependent sex determination in reptiles. We also encourage more studies on TSD patterns for different populations and species, to help improve our understanding of this fascinating phenomenon.

Supplemental Information

Supplemental Information 1 Rmarkdown supplementary file with R analysis

Click here for additional data file.

Supplemental Information 2 Description of the new flexit model

Click here for additional data file.

Supplemental Information 3 R code used for analysis

Click here for additional data file.

Additional Information and Declarations

Competing Interests

Author Contributions

Data Availability

The authors declare there are no competing interests.

F. Alberto Abreu-Grobois conceived and designed the experiments, performed the experiments, prepared figures and/or tables, authored or reviewed drafts of the paper, tested R code, and approved the final draft.

B. Alejandra Morales-Mérida and Erik Navarro conceived and designed the experiments, performed the experiments, authored or reviewed drafts of the paper, and approved the final draft.

Catherine E. Hart conceived and designed the experiments, authored or reviewed drafts of the paper, and approved the final draft.

Jean-Michel Guillon conceived and designed the experiments, performed the experiments, prepared figures and/or tables, authored or reviewed drafts of the paper, and approved the final draft.

Matthew H. Godfrey conceived and designed the experiments, prepared figures and/or tables, authored or reviewed drafts of the paper, and approved the final draft.

Marc Girondot conceived and designed the experiments, performed the experiments, analyzed the data, prepared figures and/or tables, authored or reviewed drafts of the paper, and approved the final draft.

The following information was supplied regarding data availability:

Raw data is available within the R package embryogrowth, https://cran.r-project.org/web/packages/embryogrowth/index.html

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
