# Peer review of "Recent advances on the estimation of the thermal reaction norm for sex ratios"

_PeerJ, doi:10.7717/peerj.8451_

## Round 0.1 · original submission · Major Revisions

Two reviewers and myself have now reviewed your manuscript. I believe all three of us are in agreement that your manuscript is not only a well-done analysis but also a well-written study. Most of the comments from myself and the reviewers pertain to text changes and citations, although there are suggestions some limited additional analysis that could improve this manuscript.

Reviewer 1 highlights three key issues in their ‘Basic Reporting’ section; I agree with each of these points and they all must be addressed. Reviewer 1 also suggests an additional dataset you could use to analyze and compare to the current analyzed data. While this is, of course, not a requisite for publishing your study, it would likely be a useful demonstration of how your method performs on data of varying quality and quantity. Reviewer 2 mentioned that the manuscript may not fall within the Aims and Scope of PeerJ. I understand where the reviewer is coming from; from my perspective, though, your manuscript falls within the scope of a research article, particularly because you compare multiple methods and apply the method to existing data. That being said, I would encourage the authors to perhaps modify the final paragraph of their introduction to not only describe your study as providing a “step-by-step workflow” but also one which compares methods and applies the methods to a species which is IUCN-listed as “Vulnerable”

I also have a few comments of my own that need to be addressed. The authors must remove all derivations of the term “feminization” or "feminizing" from the study. It is not only biologically inaccurate but derogatory to many people and represents exclusionary rhetoric. For example, instead of using “over-feminization” perhaps use “an abnormally high production of females”. I have written on this previously (see: https://www.washingtonpost.com/outlook/2019/06/10/how-gendered-language-leads-scientists-astray/?noredirect=on) as have other scholars; it would be best to amend this language.

When you discuss sex ratios as a binomial rather than a Gaussian response (~ line 213), I think it would be useful to describe why this is. Many readers may not be familiar with treating sex ratios (or really any ratio) as a binomial response despite it being statistically sounder to do so. Given the eloquent statistical discussions throughout your manuscript, I think it would be important to have that discussion here as well. This distinction actually matters given a binomial response should account better for sample size variation across the temperature gradient.

There are some instances throughout that require further citations. For instance, under “Fitting a model to the data: maximum likelihood” there is a discussion of intersex individuals and even phrases like “Based on the current knowledge”. Yet there are no citations for these sentences.

You remove the intersexes from your study because they only represented ~4% of the data. However, I find it interesting that the two studies finding intersexes were Costa Rica and Mexico. Does this perhaps suggest local conditions in this part of the Pacific that differ from Brazil’s Atlantic coast or India? I wonder if your manuscript could be helpful to include a multinomial model for this particular subset of data to illustrate the use of the multinomial model? Given your results suggest a simple model is sufficient for these particular data, I’d be curious if a multinomial model would be interesting or not?

Lines 46-47: perhaps indicate the sexual phenotype is “exclusively” defined by temperature in the absence of a genetic basis of phenotypic sex. This would help distinguish from thermal sex reversal.

Lines 512-513: please refrain from strong language like “which is obviously wrong” here and throughout the manuscript.

For Tables 2 and 3, please define the header terms in the figure legend. It will be easier for readers to not have to go back to the main text to look them up.

I look forward to a revised version of your manuscript and congratulate you on a well-done study.

·

Basic reporting

In this manuscript, the authors propose a new method to estimate the temperature-sex reaction norm at constant temperature for reptile embryos with TSD. They provide a flexible model for estimate the temperature-sex reaction norm using maximum likelihood, as well as a method of estimating the temperature-sex reaction norm using a Bayesian method. They include clear, step-by-step instructions aimed at reducing barriers for authors analyzing TSD data.

Overall, I enjoyed the manuscript and believe it will be useful to future researchers, in terms of the clarity of methodological explanations, the new flexit model they propose which is superior to the commonly used logistic model, as well as the Bayesian MCMC method for model fitting. However, there are some small issues with language, formatting, and references (described herein, and in “General Comments”), as well as also some larger issues with regards to the data used for their analyses (see under “Validity of Findings”, that must be addressed before Acceptance.

1. The manuscript is written in a logical and easy-to-follow way, and is useful in nature. However, the quality of the language used is at times informal (for example, Line 264, see elaboration under “General Comments”) and appears unfinished due to the use of bullet points (see Line 253-258 and elsewhere, under “General Comments”). Further, the language used is inconsistent; the authors state that they will use “TSD pattern” to describe the temperature-sex reaction norm on line 66, but afterwards continue to refer to the reaction norm (see “General Comments” for exact lines). I suggest the manuscript be refined and edited, in terms of language, before Acceptance.

2. The authors provide an excellent general overview of the TSD phenomenon in their introduction. That being said, the thermosensitive period and its complexities should be discussed in more detail; the middle-third of incubation, though commonly used, is a poor proxy for the TSP for several reasons – the authors hint at this, but do not describe it in detail, and lack references to key papers describing the relationship between thermosensitive period, temperature, and development rate (see “General Comments” under lines 115 – 116, Line 516 for more detail). Accurately delineating the thermosensitive period is key to estimating sex ratios (whether at constant, or fluctuating temperatures) and this should be stressed, especially given the usefulness of Girondot’s embryo growth package, which integrates development rate with temperature.

3. Figures and tables could benefit from further description; I found the captions somewhat vague. Further, they should be integrated into the text more comprehensively. For example, on lines 484, the authors state that comparisons of fit using Maximum Likelihood and Bayesian MCMC are shown in Table 5, yet the Maximum Likelihood estimates are not shown in Table 5. In Table 2, the caption states “bold lines indicates the selected model”; there are no bold lines in Table 2. In table 4, the caption states “selected models are indicated in bold font”, but there is no bold font. I also suggest the authors elaborate on their table and figure captions to include Species and Locations.

Experimental design

No comment.

Validity of the findings

1. While I find the methods the authors have used intriguing, and understand that the purpose of the paper is to present these new methods, I found the source data they were using to be low-quality when compared to other datasets in the literature being used to calculate the temperature-sex reaction norm in turtles. See, for example, Wibbels et al 1998 (reference below this section), where sample sizes were 19 or greater for each temperature, in this species. For example, at every temperature in Pirambu Beach and Orissa, as well as the majority of temperatures in Playa La Destiladeras, there are fewer than 10 embryos used. At 26.3C in Pirambu beach, only two embryos were sexed; I do not believe that, with certainty, you could say this temperature produces all males. I understand that sample sizes can be low for protected species (as you said on line 340) but these sample sizes are extremely low. I give two suggestions, either of which would improve the quality of the paper:

a) Give significant justification and explanation for the sample sizes used and explain to the reader how this would affect analyses (Bayesian MCMC vs. MLE).

b) If possible, could you use and/or compare this with another dataset, perhaps from another species in which there are high sample size estimates of the temperature-sex reaction norm, to see how your new method compares with MLE?

2. Within your text, could you justify the pooling of data from various sites? For example, Nancite and El Verde Camacho are over 2500km apart as the crow flies, I am not sure why it makes sense to pool these data; in a study of freshwater turtles, Ewert (2005; reference below) found significant differences in temperature-sex reaction norm between populations that were closer geographically.

3. Relating to #2 (above) - La Escobilla and Playa La Destiladeras have very few temperatures represented. Note that in Table 1, La Escobilla has 32C represented twice (perhaps a typo?), giving a total of three temperatures. Please clean up or clarify this table. Can you justify the use of sites with four or fewer temperature treatments?

3. Concomitant with comment #2 under Basic Reporting, proper delineation of the TSP is very important to accurately estimating sex ratios (even at constant temperature) under TSD. As we know, temperatures can fluctuate to a high degree (relative to the TRT) within an incubator. Is it possible to modify your model with an understanding of the TSP in L. olivacea from stage-shift experiments, increasing the resolution from simply "middle third of incubation" (E.g. Massey et al 2019 for the Snapping Turtle)? This is just a suggestion, I would be curious to see how it affects your modeling of the temperature-sex reaction norm.

References:
Ewert, M. A., Lang, J. W., & Nelson, C. E. (2005). Geographic variation in the pattern of temperature-dependent sex determination in the American snapping turtle (Chelydra serpentina). Journal of Zoology, 265(1), 81–95. doi:10.1017/s0952836904006120

Wibbels, T., Rostal, D., & Byles, R. (1998). High pivotal temperature in the sex determination of the olive ridley sea turtle, Lepidochelys olivacea, from Playa Nancite, Costa Rica. Copeia, 1998(4), 1086-1088.

Additional comments

Line 46: It may be more comprehensive to mention that some fish exhibit TSD (see: Ospina-Alvarez, N., & Piferrer, F. (2008). Temperature-dependent sex determination in fish revisited: prevalence, a single sex ratio response pattern, and possible effects of climate change. PloS one, 3(7), e2837.)

Line 47: “termed” the “thermosensitive” period. Termed is preferable to named as “thermosensitive period” is jargon. Thermosensitive rather than thermo-sensitive has been used elsewhere in the manuscript and is more commonly used in the literature.

Line 51: Present in some turtles is more accurate. Snapping Turtles, for example, have Type II, whereas many softshell turtles have GSD.

Lines 72-74: I am confused by this statement; if latitude (local incubation conditions) explained differences in TRT breadth, why do geography (which encapsulates latitude) and local thermal conditions (which encapsulates local incubation conditions) not explain variation in TRT breadth?

Line 81: I recommend changing pertinent to empirical.

Line 92 – 93: The citation format here and elsewhere are incorrect.

Line 113: “The” second-third. Suggestion: many authors refer to it as the middle third.

Line 115 – 116: The assumption that the middle third of incubation corresponds exactly to the TSP is likely invalid in general, and likely varies with temperature; see several stage-shift experiments, see:

Merchant-Larios, H., Ruiz-Ramirez, S., Moreno-Mendoza, N., & Marmolejo-Valencia, A. (1997). Correlation among Thermosensitive Period, Estradiol Response, and Gonad Differentiation in the Sea Turtle Lepidochelys olivacea. General and Comparative Endocrinology, 107(3), 373–385. doi:10.1006/gcen.1997.6946

Yntema, C. L. (1979). Temperature levels and periods of sex determination during incubation of eggs of Chelydra serpentina. J. Morphol. 159, 17–28.

Line 125: it may be useful to provide citations of studies that have shown mean temperature is a poor proxy for TSD patterns at constant temperature. Reword as: “The use of mean incubation temperatures obtained from natural nests to feed a model to estimate TSD patterns has been shown to produce inaccurate results.”

Line 126: “A further misconception is the difference between resolution…”

Line 143: It is unclear whether you are using data from nesting beaches (in-situ) or are using common-garden data (incubated in lab at constant conditions). Please clarify.

Lines 146 – 148: Embryo growth reaction norms from different locations are highly likely to differ. Did you estimate different embryo growth reaction norms for all populations, or just Pacific Mexico vs. Pacific Costa Rica?

Line 152 and elsewhere: The bullet points make the manuscript difficult to read and look unfinished; can these be sectioned into paragraphs or subheadings rather than bullet points?

Line 180: Can you elaborate on how the Hulin model is challenging to fit?

Line 220: Pieau published this work in 1998 in the Journal of Experimental Biology; it would be useful to cite this paper for other authors.

Line 242: “To fit the TSD pattern using maximum likelihood…” (Delete unnecessary words; previously, you state that you will use “TSD pattern” to describe temperature-sex reaction norm – keep your wording consistent.)

Line 252: “serves various purposes”

Lines 253 – 258: I suggest removing the bullet points and replacing this with a paragraph for clarity and flow. Further, please explain each point in more detail for the reader – e.g. how is it that datasets become comparable?

Line 264: Language is informal and not cohesive with the rest of the manuscript (“if you slightly shift a parameter”…) Consider changing to passive voice: “if a parameter is slightly shifted” to make the paragraph more cohesive.

Line 286: Same as above. Change to “In short, if the standard error of the maximum likelihood parameters are known…”

Line 326: Akaike (1973) is the citation for the original concept.

Lines 372 – 375: I suggest numbering these bullet points and/or giving the symbol for each parameter for readability.

Lines 381 – 382: I suggest removing these bullet points and rephrasing them into a paragraph for cohesion and to add a finished quality.

Line 496: I suggest changing “original methods used were rather crude” to something less harsh; the advanced tools we have now were not necessarily available to authors at the time. Perhaps state “significant methodological advancements have been made since early estimates using…” or something similar.

Line 516: Please include other references here; other authors have noted that nonlinear development rates affect duration and onset of TSP (Georges, 2005); further, others have noted that incubation temperature affects variation in the TSP (Merchant-Larois et al., 1997; Yntema, 1979).

Line 520: The citation indicating that large-scale empirical validation is lacking came before the studies for which you cite improved models. For example, Massey et al. (2019) and Carter et al. (2019) both had extremely large empirical data sets. I would recommend changing your last sentence to something like, “but further analyses of in-situ empirical data would benefit the field.”

Line 522: I disagree that existing statistical methods have inhibited authors analyses and study of TSD; for example, in preparation for Massey et al. (2019), several models were tested (Hill, alpha-logistic, etc.), all of which produced similar results – this did not cause inhibition in any way. However, I enthusiastically agree that in this paper, the R code provided will simplify the process of analyses for authors. I would re-word this paragraph to affirm that a lack of accessible methodology creates “barriers” for authors, and highlight your contribution thereafter.

Lines 529 – 542: I recommend placing this in paragraph form for flow and cohesion.

Line 541: Switch “think” to “suggest”.

Reviewer 2 ·

Basic reporting

This manuscript presents a detailed explanation of one method for fitting a function to constant-incubation data for reptiles with temperature-dependent sex determination. The writing is excellent and 99% clear, with a few locations where the wording could be improved (noted in General Comments, below). Figure 5 would be better if presented with a different color scheme that is more friendly for grayscale viewing and colorblind readers.

Experimental design

This paper reports on the various modeling/statistical methods available in an R package, using sea turtle data in a worked example. It does not pose or answer a scientific question, per se. As such, the editors should determine whether this manuscript fits within the Aims & Scope of the journal or would be considered a case study. If necessary, the paper could be re-worked relatively easily to focus more on the ecological question of TSD in olive ridleys and less on the statistical methods.

That being said, this paper is an excellent treatment of its subject and much needed in the TSD modeling community. The statistical descriptions are detailed, rigorous, and clear for the non-statistician. One could reasonably argue that the detailed explanations of general methods, like AIC, are not strictly needed, since they have been described elsewhere, sometimes to excess. However, my opinion is that it's useful for reference purposes to have all of the relevant models and stats in a single paper, and I would favor keeping them in.

Validity of the findings

Data and data sources are provided, along with R code for running the example. The R functions run as expected.

The authors do state conclusions about the various turtle rookeries, but they are really just an afterthought to the methodological focus. Again, though, this section could be re-worked if the manuscript needs to be more question-focused to fit the journal.

Additional comments

I have some minor comments, so I added line numbers to the manuscript for ease of referencing.

Since this paper focuses on methods for fitting TSD data, not just for sea turtles, I think the title would be improved by removing taxon references.

L97-103: Lists of arguments are difficult to read. Consider putting them in a more organized format (table, etc.).

L295: Consider replacing 'high volume' with 'many' because the former has several definitions that could be semantically confusing.

L334: Different authors have presented different perspectives on the delta-AIC value that should be considered equivalent (from 2 to at least 6). It would be good to cite at least one of those perspectives here or, alternatively, note the justification for the choice of 2 as a delta-AIC cutoff in this paper.

L337: Elaboration of this point would be beneficial, as well as justification for the Hurvich-Tsai formula. It would also be useful to note that the AIC value itself is not strictly informative in terms of absolute model fit.

L343: It's not atypical for common species, either, which might be even more important to note.

L358: It's not 100% clear what a ‘dataset’ would be in this case. Results from multiple incubation studies with the same species? All studies that use same/different set of incubation temps? Etc.

L397: If you’re already starting with max likelihood values, it would be reasonable to assume that your estimates aren’t complete nonsense (otherwise, no method would be trustworthy). In that case, why would you also need to burn in your chain? See, e.g., the Geyser essay in the Brooks et al. 'Handbook of MCMC'.

L411: Shift data description to the Methods section.

L427: Given how well-documented the rest of the stats methods are, a quick reference to where the 98% value comes from would be useful here.

L442-460: Much of this paragraph is repeated from the Methods section and is unnecessary here.

L519: The results from this paper don't seem to indicate that an incorrect TSD pattern is a likely explanation for not detecting substantial variation among populations.

L520: Provide an example of this being done in the literature.

---

## Round 0.2 · Minor Revisions

I appreciate your thorough responses to the reviewers and find your manuscript to be greatly improved. I have only several small comments for you to include before I can accept this manuscript.

Change “intersexuality” to “intersex” throughout
Line 88: IUCN is misspelled
Line 90: be more clear as to who “them” is

Previously one of the reviewers made the below comment. In your rebuttal you disagreed but did not explain why you disagreed. I believe you should provide a more thorough explanation of this in your manuscript.

"Accurately delineating the thermosensitive period is key to estimating sex ratios (whether at constant, or fluctuating temperatures) and this should be stressed, especially given the usefulness of Girondot’s embryo growth package, which integrates development rate with temperature."

Thank you for resubmitting this improved revision and I look forward to the next version of your manuscript.

---

## Round 0.3 · accepted · Accept

Thank you for revising your manuscript, it is greatly improved and will be a useful contribution.